# Composite Proton Exchange Membranes Based on Chitosan and Phosphotungstic Acid Immobilized One-Dimensional Attapulgite for Direct Methanol Fuel Cells

**DOI:** 10.3390/nano10091641

**Published:** 2020-08-21

**Authors:** Wen-Chin Tsen

**Affiliations:** Department of Fashion and Design, Lee-Ming Institute of Technology, New Taipei City 243, Taiwan; charity@mail.lit.edu.tw; Tel.: +886-02-29097811

**Keywords:** chitosan, attapulgite, phosphotungstic acid, proton exchange membranes, fuel cells

## Abstract

In order to obtain biopolymer chitosan-based proton exchange membranes with excellent mechanical properties as well as high ionic conductivity at the same time, natural attapulgite (AT) with one-dimensional (1D) structure was loaded with a strong heteropolyacid and also a super proton conductor, phosphotungstic acid (PWA), using a facial method. The obtained PWA anchored attapulgite (WQAT) was then doped into the chitosan matrix to prepare a series of Chitosan (CS)/WQAT composite membranes. The PWA coating could improve the dispersion and interfacial bonding between the nano-additive and polymer matrix, thus increasing the mechanical strength. Moreover, the ultra-strong proton conduction ability of PWA together with the interaction between positively charged CS chains and negatively charged PWA can construct effective proton transport channels with the help of 1D AT. The proton conductivity of the composite membrane (4 wt.% WQAT loading) reached 35.3 mS cm^−1^ at 80 °C, which was 31.8% higher than that of the pure CS membrane. Moreover, due to the decreased methanol permeability and increased conductivity, the composite membrane with 4% WQAT content exhibited a peak power density of 70.26 mW cm^−2^ fed at 2 M methanol, whereas the pure CS membrane displayed only 40.08 mW cm^−2^.

## 1. Introduction

Direct methanol fuel cell (DMFC) is a highly efficient and eco-friendly energy conversion device that can directly convert chemical energy into electrical energy by using liquid methanol as the anode fuel. Moreover, when compared with proton exchange membrane fuel cells (PEMFCs) using gaseous hydrogen as the fuel, DMFCs have a wider application prospect especially in the fields of portable devices and vehicles because they can fully utilize the existing infrastructure for gasoline [1,2,3]. As a key component in a DMFC, the proton exchange membrane (PEM) plays an important role of a separator of anode and cathode as well as a deliverer of protons [4,5]. PEMs must possess high proton conductivity, excellent methanol barrier ability, and good thermal and mechanical properties. Up to now, perfluorosulfonic acid (PFSA) membranes (represented by Dupont’s Nafion) are known as currently commercial PEMs in PEMFCs for their relatively high proton conductivity and good chemical and electrochemical stabilities [6]. However, the high methanol permeability of PFSAs is a big obstacle when utilizing PFSAs as PEMs in DMFCs. In addition, the very high cost and limited operation temperature of PFSAs also bring adverse effects on their widespread application in DMFCs [7]. Therefore, numerous researchers have turned to seek cost-effective and high-performance alternatives to PFSAs.

In recent years, chitosan (CS), a natural polymer, has attracted wide attention as a PEM material for its abundance in nature, environmental friendliness and low cost [8,9]. As the only degradable polycationic bio-polysaccharide in nature, CS is a deacetylated product of chitin and has been applied to various fields such as food chemistry, biomedicine, cosmetics, textile, papermaking and so on. CS has a large number of amino and hydroxyl groups, and good ability of alcohol–water separation, which has preferential permeability to water under the condition of alcohol resistance. So, CS is also a good candidate of PEMs to solve the problem of methanol permeation [8]. In the process of membrane preparation, the hydrophilic groups of CS generate strong intra molecular and intermolecular hydrogen-bonding interactions, which drive CS chains to form numerous crystalline regions and cross-linked networks. On the one hand, the high crystallinity packs the CS chains and thus decreases the free volume cavities (only 0.56 nm), which can effectively prevent methanol penetration through the CS membrane [10], but on the other hand highly crystallized CS has very low proton conductivity because the ion conduction in a PEM mainly occurs in the amorphous phase rather than crystalline phase [8,9]. Additionally, the poor mechanical properties of CS are also a concern when using CS-based PEMs operation in fuel cells. To improve the mechanical properties and proton conductivity, many efforts have been taken to modify CS. For example, chemical modification [11,12] (e.g., sulfonation, phosphorylation, chemical cross-linking) and physical modification [13,14,15,16] (e.g., blending with other polymers, organic-inorganic hybrids). Among these methods, preparation of organic-inorganic composites is considered as a facile and efficient approach to solve the problems facing CS because this method can combine both their merits and sometimes generate the synergistic effect.

Many inorganic nano-additives such as 0-dimensional (0D) nanoparticles [17,18,19] (e.g., silica, titania, zirconia, zeolites, silicon-aluminum oxides), one-dimensional (1D) nanotubes or nanorods [14,15,16,20,21,22] (e.g., carbon nanotubes, halloysite tubes) and two-dimensional (2D) nanoplates [23,24] (e.g., montmorillonite, grapheme oxide) have been extensively applied to modify polymers to improve their thermal and mechanical stabilities while endowing them with some new properties due to the synergistic effect. Among those inorganic nanomaterials, 1D nanotubes or nanorods stand out for their unique anisotropic 1D shape because such nanomaterials can contribute to construct better channel-like ion transport pathways in the composites for proton conduction [25,26,27]. Generally, nanomaterials including 1D nano-additives need to be surface modified to improve the compatibility between nanomaterials and polymer matrix, and thus, can fully play their functions. Wen and co-workers designed and prepared functionalized carbon nanotubes (CNTs) with different surface coating substances (such as chitosan, sulfate zirconia, SiO_2_, TiO_2_, organic long-chain ions) to modify CS to fabricate a series of composite membranes [14,15,16,26,28]. These surface coating materials can not only promote the dispersion of CNTs, and thus, fully play the reinforcement role of CNTs, but also improve the proton conductivity due to the newly formed proton conducting network along the surface of functionalized CNTs. Apart from 1D CNTs, 1D clay nanotubes or nanorods have also been used as an effective additive in the field of PEMs. Wang et al. [21] synthesized halloysite nanotubes bearing sulfonated polyelectrolyte brushes (SHNTs) via distillation-precipitation polymerization and then added into CS matrix to prepare nanohybrid membranes. The SHNTs is helpful to create continuous pathways along the nanotube, thus generating acid-base pairs at SHNT-CS interface, which can work as low-barrier proton-hoping sites. Moreover, the wider pathways could form with the aid of the long brushes on SHNTs in the CS matrix.

Recently, phosphotungstic acid (PWA), recognized as a strong heteropolyacid and a super proton conductor, was immobilized onto the surface of poly (vinylidene fluoride) (PVDF) electrospun nanofibers to obtain a new three-dimensional proton conducting network [29]. After impregnating with CS, which can tightly anchor PWA to avoid its leaching out from PEMs in water or methanol, the resulting composite membrane demonstrated ultrahigh proton conductivity and satisfactory fuel cell performance. Inspired by that, herein, we utilized PWA as a surface modification material to coat 1D natural clay attapulgite (AT) with a theoretical formula of Si_8_Mg_5_O_20_(OH)_2_(H_2_O)_4_·4H_2_O [30]. It should be mentioned that AT is cheap and abundant in nature when compared with other carbon nanomaterials [23,25]. Then, the obtained PWA-decorated AT was added into the CS matrix to fabricate novel organic-inorganic composite membranes. Expectedly, the PWA coating could improve the dispersion and interfacial bonding between the nano-additive and polymer matrix, thus increasing the mechanical strength. Moreover, the ultra-strong proton conduction ability of PWA together with the interaction between positively charged CS chains and negatively charged PWA can construct effective proton transport channels with the help of 1D AT. The effects of PWA-coated AT on the structure and properties of CS were evaluated. The single direct methanol fuel cell performance was also tested.

## 2. Materials and Methods

### 2.1. Materials

Attapulgite clay was obtained from the Huaiyuan Mining Industry Co. (Huai’an, Jiangsu Province, China). Chitosan (*M*_w_ = 500 kDa) with a degree of 90% deacetylation was purchased from Zhejiang Aoxing Biotechnology CO., Ltd. (Yuhuan, Zhejiang Province, China), DC5700 [(CH_3_O)_3_Si(CH_2_)_3_N^+^(CH_3_)_2_(C_18_H_37_)Cl^−^] was bought from Green Chem. International Co., Ltd. (Shanghai, China). BaCl_2_ was purchased from Annaiji Chemical Agent CO., Ltd. (Beijing, China). Ethanol (A.R.), acetic acid (A.R.), PWA (A.R.), and ammonia (A.R.) were supplied by Sinopharm Chemical Reagent Co., Ltd. (Shanghai, China). Sulfuric acid (98%) was supplied by Kaifeng Shenma Group Co., Ltd. (Kaifeng, China).

### 2.2. Synthesis of Phosphotungstic Acid Immobilized Attapulgite

Phosphotungstic acid immobilized on attapulgite (WQAT) was prepared using the following main three steps: Firstly, AT was ultrasonically dispersed in a H_2_SO_4_ solution (2 M) at 60 °C for 2 h. The suspension was filtered, followed by rinsing with deionized water until SO_4_^2^^−^ was undetectable by a BaCl_2_ solution. The obtained sample was dried and gained acid-activated AT. Secondly, 2 g of acid-activated AT was dispersed into the mixture of ethanol and deionized water (1:1, volume ratio), and ultrasonically oscillated for 30 min. After adjusting the pH to 10–12 using NaOH solution, 15 mL of DC5700 was added to the mixture under ultrasonic dispersing at 70 °C for 120 min, and then slowly stirred for 12 h. The mixture was filtered, repeatedly washed by water and dried at 80 °C to obtain quaternized AT (QAT). Finally, WQAT was obtained by immersing QAT in a PWA solution (5 mg mL^−1^) for 24 h and then washed by deionized water to remove the residual PWA, and dried in a vacuum oven at 60 °C overnight.

### 2.3. Preparation of CS-Based Composite Membranes

An amount of 0.7 g of CS was dissolved in a 20 mL of acetic acid aqueous solution (2% *v/v*) and stirred at room temperature. Simultaneously, a certain amount of nanoparticle was dispersed into a 10 mL of acetic acid solution with ultrasonic treatment for 30 min. Then, these two solutions were mixed together and stirred vigorously for another 24 h. The resultant homogenous mixture was cast onto a clear glass plate and dried at 40 °C for 24 h to obtain composite membranes. To easily uncover the membranes and remove the remained acetic acid, the membrane was immersed in a NaOH solution for 2 h and then rinsed with deionized water until the pH value was 7. Subsequently, the membrane was ionic cross-linked by immersion in a H_2_SO_4_ solution (2 M) for 24 h and then extensively washed with water to remove the residual H_2_SO_4_. Finally, the membrane was dried under vacuum at 30 °C for 24 h. The obtained composite membranes are named as CS/WQAT-x, where x is the mass percent of the WQAT in the CS matrix. The average thickness of the dry membranes falls in the range of 50–55 μm.

### 2.4. Characterization

Nicolet 380 Fourier transform infrared spectrometer (Thermo Electron Co., Waltham, MA, USA) and a X-ray photoelectron spectroscopy (XPS, ThermoFisher Scitific Co., Waltham, MA, USA) were used to character the chemical structure of AT and modified AT samples. The thermal properties of WQAT and composite membranes were conducted by thermo gravimetric analysis (TGA). TGA was carried out on a STA 499F instrument (NETZSCH Co., Germany) from 30 to 800 °C with a heating rate of 10 °C min^−1^ under nitrogen atmosphere after being kept for 5 min at 130 °C to remove the absorbed water. The thermal transition behavior of the membranes was tested on a differential scanning calorimeter (DSC, NETZSCH Co., Selb, Germany). The samples were first preheated from room temperature to 130 °C with 10 °C min^−1^ under nitrogen atmosphere, then cooled to 90 °C, and reheated to 260 °C. In order to verify the dispersion of nanorods in the composite membranes, Scanning Electron Microscope (JSM-6510, Electronics Co., Fukuoka, Japan) was used to observe the surface and cross-sections of the membranes at 5 kV. The membrane sample was freeze-fractured in liquid nitrogen previously and then sputtered with gold. The mechanical properties of the hybrid membranes (1.0 × 4.0 cm) were investigated by a universal tensile testing machine (AG-IC 5KN, Shimadzu Co., Kyoto, Japan) using an elongation rate of 2 mm min^−1^ at room temperature.

Water uptake and swelling ratio of the membranes were carried out through measuring the changes in the weight and area of the samples in dry and wet conditions. After weighing the weight (W_dry_) and recording the size (A_dry_), the dry membrane was soaked in deionized water for 24 h at 80 °C and then recorded the weight (W_wet_) and size (A_wet_) of the wet sample. Water uptake and swelling ratio of the membrane are calculated according to the following equations:(1)$$\mathrm{Water}\;\mathrm{uptake}\;\left(\%\right)=\frac{W_{\mathrm{wet}}-W_{\mathrm{dry}}}{W_{\mathrm{dry}}}\times 100\%$$
(2)$$\mathrm{Area}\;\mathrm{swelling}\left(\%\right)=\frac{A_{\mathrm{wet}}-A_{\mathrm{dry}}}{A_{\mathrm{dry}}}\times 100\%$$

Proton conductivity was obtained by electrochemical impedance spectroscopy (EIS) on an electrochemical station (Autolab PGSTAT 302N, Netherland) with voltage amplitude of 5 mV in the frequency range of 1 Hz–1 MHz. Initially, all the membranes were fully hydrated by immersing in deionized water for 24 h, and then put onto a home-made two-electrode mould with platinum wires as electrodes. After that, the measurements were carried out in the temperature range of 20–80 °C. The calculation formula of proton conductivity (σ, S cm^−1^) is as follows:(3)$$\sigma =\frac{L}{A\times R}$$
where L (2.16 cm) is the distance between the two electrodes, and R and A are the resistance (Ω) and cross-sectional area (cm^2^) of the membrane sample, respectively.

### 2.5. DMFC Performance and Methanol Crossover Test

The DMFC single cell performance was evaluated at 70 °C with an active area of 4 cm^2^ (2 cm × 2 cm). The anode and cathode catalysts were Pt-Ru with a loading of 4.0 mg cm^−2^ and Pt of 2.0 mg cm^−2^, respectively. The membrane was sandwiched between the anode and cathode, and then hot pressed for 3 min at 125 °C and 2 MPa to obtain a membrane electrode assembly. Methanol solution (1 mol L^−1^ H_2_SO_4_ solution in different ratio) was fed as a fuel into the anode at a flow rate of 1 mL min^−1^ by a peristaltic pump, and oxygen was fed into the cathode at a flow rate of 90 mL min^−1^. The current density (I) and potential (V) of the fuel cells were recorded by an electrochemical workstation (Autolab PGSTAT 302N, Herisau, Switzerland) at a scan rate of 5 mA s^−1^.

Methanol crossover was tested using a voltammetric method. A methanol solution (1, 2 or 5 M) was fed at a flow rate of 1.0 mL min^−1^ into the anode side of the MEA while the cathode side was kept in an inert N_2_ atmosphere at a flow rate of 90 mL min^−1^. The whole test setup was kept at a temperature of 70 °C. By applying a positive potential at the cathode side, the flux rate of permeating methanol was determined by measuring the steady-state limiting current density resulting from the complete electro-oxidation at the membrane/P_t_ catalyst interface at the cathode side.

## 3. Results and Discussion

### 3.1. Synthesis and Characterization of WQAT

The synthesis process of WQAT could be divided into three steps as illustrated in Scheme 1: The first step is a conventional acid treatment, which can introduce some hydroxyl groups on the surface of AT. With the aid of these introduced -OH groups, DC5700, a long-chain silane coupling agent with quaternary ammonium groups, grafted onto the surface of AT to obtain QAT through typical hydrolysis and condensation reactions. In the following step, with the AT nanorods and the surface grafted quaternary ammonium groups acting as a template and anchoring sites respectively, PWA, a heteropolyacid with strong acidity and have excellent proton transport ability, which can immobilize on AT through a strong acid-base interaction between positively charged QAT and negatively charged PWA.

FTIR, TG, and XPS were used to characterize the synthesized WQAT powders. The chemical composition of pristine AT and modified AT was confirmed by FTIR as shown in Figure 1a. As for the FTIR spectra of AT, QAT, and WAQT, the three samples showed the characteristic absorption bands of attapulgite at 3400–3600 cm^−1^, which correspond to the four hydroxyl structures of the hydroxyl stretching vibration of bonds to aluminum and/or magnesium, the hydroxyl stretching vibration of adsorbed water, and the bending vibration of zeolite water in the channels of AT [31]. The absorption peaks at 1030 and 1018 cm^−1^ were characteristic peaks of symmetric and asymmetric stretching vibration of Si-O of Si-O-Si and Si-O-Al, respectively. Besides, the FTIR spectrum of QAT appeared two new absorption peaks at 2924.1 and 2853.0 cm^−1^, which corresponded to the C-H stretching vibration of CH_3_ and CH_2_ in the alkyl chain of DC5700, indicating that AT was successfully modified by DC5700. The characteristic IR bands of HPW in WQAT were somewhat different from those of bulk HPW. Among the four characteristic bands of HPW, those at 1080 cm^−1^ (P-O_a_) and 981.4 cm^−1^ (W-O_d_) were overlapped by the strong and broad Si-O-Si band and Si-O band of AT, while those at 890.3 cm^−1^ (W-O_b_-W, corner-sharing) and 798.8 cm^−1^ (W-O_c_ -W, edge-sharing) shifted to 897.1 cm^−1^ and 817.0 cm^−1^, respectively. These shifts are probably assigned to the formation of the secondary structure of phosphotungstic acid with a strong chemical interaction between oxygen terminated of heteropolyacid polyanion and H_5_O_2_^+^ of QAT, which illustrated HPW has a good stability on the surface of QAT [32].

XPS was used to verify the elements (such as Al 2p, Si 2p, Si 2s, C 1s, O 1s, and others) presented on the surface of AT. From the XPS spectrum of QAT (Figure 1b), it can be found that a new N1s peak appeared at 401.86 eV, which was assigned to the quaternary amine groups from the grafted DC5700 chains, indicating that DC5700 long chains containing -NR_3_^+^ groups were successfully grafted onto the surface of AT through the reaction between the ethoxy group of DC5700 and the hydroxyl groups of AT. For WQAT, except for the peaks similar to QAT, a new peak of W4f at a binding energy of 35.34 eV became remarkably visible. The result also proved that HPW was anchored onto the surface of DC5700 modified AT through the strong electrostatic interaction between -NR_3_^+^ cations of grafted DC5700 and PW_12_O_40_^3−^ anions of PWA.

To determine the coating content of PWA in WQA, TGA was carried out in a nitrogen flow from 30 to 800 °C. The obtained TGA and derivative thermo-gravimetric (DTG, dW/dT) curves are shown in Figure 2a,b. As for the pristine AT, it showed a three weight loss thermal degradation behavior: The first loss is due to the adsorbed water and zeolite water in the attapulgite; the second loss comes from the removal of structural water; the third loss is because of the destruction of the internal hydroxyl groups of AT. As for QAT, the first weight loss (about 4.06%) was lower than that of AT (9.86%), which was mainly attributed to the hydrophobic organic long chains covering the surface of AT. Apart from this, QAT had a weight loss of 12.24% in the range of 150–270 °C and 29.09% in the range of 280–600 °C, which should be due to the introduction of a new bond of Si-O-Si between organo-silane coupling agent and AT and the decomposition of DC5700 organic molecular chains. After PWA coating, WQAT exhibited a similar decomposition procedure to that of QAT. From the difference of the final residual content between QAT and WQAT, the loading content of PWA could be calculated to be 11.48%, because PWA can keep thermally stable until 700 °C [33].

### 3.2. Characterization of Pure CS and CS/WQAT Composite Membranes

#### 3.2.1. Structure and Morphology

The dispersion of nanofillers in an organic-inorganic composite system can affect the load transfer from the polymer matrix to stiff inorganic nanofiller. Besides, the good distribution of 1D WQAT in the composite membranes could also contribute to form new channel-like pathways for proton transport. The cross-sectional morphology of the composite membranes was probed using Scanning Electron Microscope (SEM) as shown in Figure 3a–f. The pure CS membrane (Figure 3a) showed a homogeneous and impact morphology without obvious cracks or micro-voids. With the introduction of WQAT, which are nanorods with a diameter of ~10 nm and length of several microns (as shown in Figure 3g), obvious white particle-like substances can be observed in the composites (as shown in Figure 3b–f). These white particles are WQAT nanorods distributed in the CS matrix after liquid nitrogen quenching during the preparation of SEM cross-sectional samples. Clearly, these WQAT nanorods were uniformly dispersed in the polymer matrix as for the sample CS/WQAT-2 and CS/WQAT-4, which can be verified by the Energy Dispersive X-Ray Spectroscopy (EDX) mapping images of Si (Figure 3h) and W (Figure 3i). However, some WQAT agglomerations (marked with yellow circles in Figure 3d–f) can be seen in the cross-sections of the composite membranes when the content of WQAT was more than 4%.

Generally, the proton conduction in PEMs mainly occurs in amorphous phase rather than crystalline phase [8,9]. Due to the existence of intramolecular hydrogen bonds between oxygen atoms and intermolecular hydrogen bonds between amino and hydroxyl groups in CS molecule, CS regularly accumulates into a semi-crystalline structure during the process of film formation. The XRD patterns of the as-prepared membranes are depicted in Figure 4. The pure CS membrane exhibited typical three characteristic diffraction bands at 2θ = 12.2°, 30.8° and 40.5°. Obviously, as for the composite membranes, the intensity of the strong band of CS matrix at 12.21° decreased first and then increased with increasing WQAT contents. Among all the membranes, CS/WQAT-4 had the weakest diffraction intensity. The reason for this trend of intensity change is maybe because the ordered arrangement of CS molecular chains is partly destroyed by the added WQAT, and thus, reduced their crystalline domains. However, when the content of WAQT was high (e.g., above 4%), the agglomerated inorganic particles (as illustrated in SEM images in Figure 3) had no obvious damage ability to the crystal areas. This effect of WQAT on the crystallization ability of CS matrix may also bring some influences on the proton conductivity, which will be discussed in Section 3.2.4.

#### 3.2.2. Thermal and Mechanical Properties of the Composite Membranes

As a key component in PEMFCs or DMFCs, PEMs must have excellent thermal and mechanical stabilities for long-time operation. The thermal properties of the as-prepared membranes were investigated by TGA-DTG and DSC as shown in Figure 5. From Figure 5a, the pure CS membrane exhibited two thermal weight loss stages: (i) The first stage may be related to the degradation of chitosan side groups around 200–250 °C; (ii) the second stage in the temperature range of 250–350 °C is the decomposition of polymer backbone of CS, which is similar to the result in the literature [14]. After the incorporation of WQAT, the CS/WQAT composite membranes demonstrated similar degradation behavior to that of the pure CS membrane. By comparing the initial decomposition temperature of the first stage, the CS/WQAT composite membranes showed a slightly higher degradation temperature than that of pure CS from the DTG curves. This may be due to the interaction between phosphotungstic acid and -NH_2_ of chitosan, which partly inhibits the movement of polymer segments and thus increases the thermal stability of the composites. To further confirm the thermal behavior, DSC was also conducted from 90 to 260 °C, and the obtained curves are shown in Figure 5b. Clearly, the decomposition temperatures (T_d_) increased from 217.4 °C (pure CS membrane) to 232.2 °C (CS/WQAT-15 membrane) with the increase in WQAT content, verifying the enough thermal stability of our composite system for the application in PEMFCs and DMFCs.

PEMs should possess enough mechanical strength and adequate flexibility to meet the requirements of fuel cell assembly and operation. The tensile strength, elongation and modulus of the pure CS and CS/WQAT composite membranes are shown in Table 1. The pure CS membrane possessed a tensile strength of 41.31 MPa, elongation of 17.01% and Young’s modulus of 618.9 MPa. As for the composite membranes, the tensile strength, elongation and Young’s modulus values increased with the increase in the content of WQAT up to 4 wt.%. The improvement of tensile strength and Young’s modulus might be attributed to the good dispersion of WQAT, which contributes to the formation of more acid-base pairs, and thus, can partly prevent the slippage of CS chains when the composites are subjected to stress. The increase of elongation, which is unlike other typical inorganic nanofiller reinforced composite systems, may be ascribed to the flexible DC5700 long chains grafted on AT. Among all the composite membranes, the CS/WQAT-4 membrane exhibited the highest tensile strength, elongation and Young’s modulus of 58.65 MPa, 23.21% and 1152.69 MPa, which improved by 42.0%, 36% and 86.25% respectively when compared to those of the pure CS membrane. However, with the further increase in WQAT content, the overall decrease of mechanical properties is due to the agglomeration of WQAT, which weakens its ability of strengthening and toughening. Fortunately, the mechanical properties of the composite membranes were still better than those of the pure CS membrane. In summary, the above result showed that the incorporation of WQAT can enhance the thermal and mechanical stabilities of CS matrix, making the composite membranes more suitable for DMFCs application.

#### 3.2.3. Water Uptake and Area Swelling of the Membranes

Water molecules play an important role for proton conduction in proton exchange membranes because H^+^ ions transport generally along the hydrogen-bonded network formed by water (diffusion mechanism) or through the ion exchange groups dissociated by water (hopping mechanism). However, too high water uptake generally results in excessive swelling of PEMs and thus sharply decreases their mechanical stability. In addition, it should be mentioned that too big swelling ratio of a PEM also brings a severe methanol crossover from the anode to cathode through electro-osmosis, thus producing a mixed potential to decrease the open circuit voltage. Therefore, the water absorption and swelling of PEMs are important indicators for judging the quality of proton exchange membranes. The water uptake values of the CS/WQAT composite membranes with various weight fractions of WQAT at different temperatures are shown in Table 2. Compared with the pure membrane, the water absorption values of the composite membranes increased by 8.35% to 41.16% at 80 °C with different content WQAT. This increase in water uptake may be because AT itself has many hydroxyls on its structure, which is helpful to absorbing water. Besides, the existence of heteropolyacid anions on the surface of AT could also contain water molecules through hydrogen bonds. As for the effect of temperature on moisture uptake, all the membranes showed an increasing trend of water content with the increase in temperature. Generally, the increased temperature can accelerate the movement of water molecules and polymer chains of chitosan, which makes H_2_O easily diffuse into the membranes.

As for the area swelling of the composite membranes (as shown in Table 3), all the membranes display increased area swelling values with the testing temperatures, which is similar to the result of water uptake. However, unlike the change trend of water uptake with the content of WQAT, the area swelling values follow a slight decrease trend, indicating the increased dimensional stability of the composites. The result is probably due to the strong interfacial interaction between CS and WQAT. In summary, the above results indicated that the composite membranes possessed higher water uptake ability and dimensional stability than those of the pure CS membrane.

#### 3.2.4. Proton Conductivity, Methanol Permeability and Single Cell Performance

The proton conductivities of the as-prepared membranes were tested at 100% R.H. in the temperature range from 20 to 80 °C using a two-probe electrochemical impedance spectroscopy method, and the results are shown in Figure 6. According to the curves, the conductivity values of the composite membranes increased firstly and then decreased with the addition of WQAT. Among all the membranes, the CS/WQAT-4 membrane showed a highest conductivity of 35.3 mS cm^−1^ at 80 °C that was about 37% higher than that of the pure CS membrane (25.8 mS cm^−1^). To better understand the effect of WQAT on the proton conductivity, we also prepared the CS/AT composite membrane with 4% AT loading and tested its proton conductivity under the same condition. The result showed the proton conductivity of the CS/AT (4% AT content) composite membrane was 28.8 mS cm^−1^ at 80 °C, which was much lower than that of the CS/WQAT composite membrane with the same inorganic nanofiller content, indicating the positive influence of the PWA surface medication on AT. This significantly improved proton conductivity and may be explained as follows: (1) the reduced degree of crystallinity of chitosan matrix (as proven in Figure 5) is conducive to proton transport; (2) the super-acidic PWA coated on the surface of AT can act as new H^+^ transfer sites; (3) with the assistance of 1D structure of AT, the homogeneously dispersed WQAT (as shown in Figure 3c) could construct continuous proton transport nanopathways along the interface between CS and WQAT because of the acid-base interaction between the two components [29]; (4) the increased water uptake ability can also contribute to the fast transport of H^+^. However, it can also be noted that further increasing WQAT did not produce higher proton conductivity. For example, the CS/WQAT-15 composite membrane exhibited a conductivity of 25.8 mS cm^−1^ (80 °C) that was 73% of that of the CS/WQAT-4 membrane. Such a trend is unsurprising because the aggregated inorganic nanofiller may partly block the fast proton transportation in the composite to some extent.

In addition to proton conductivity, the penetration of methanol is another serious problem deserving to be discussed because methanol crossover can not only partly waste the anode fuel, but also affect the output voltage and fuel cell performance [34]. Considering the optimal properties of the CS/WQAT-4 composite membrane among all the composites, this sample was selected to be further evaluated for the methanol permeability and single-cell performance. The methanol crossover was tested using a voltammetric method [35], in which the limited current density that comes from complete electro-oxidation of methanol permeation at the interface of membrane/catalyst is used to evaluate the methanol permeability. Generally, the higher the methanol crossover current density, the more serious the methanol permeability. Figure 7a compares the methanol crossover current density for the hybrid membrane (CS/WQAT-4) at 70 °C in different methanol solutions (1 M, 2 M and 5 M); it can be seen that the crossover current density gradually increased with the methanol concentration. This is because methanol permeability is more serious when using a methanol solution with higher CH_3_OH concentration as the anode fuel, due to the bigger concentration gradient diffusion. At the same time, we also compared the methanol crossover current densities of the pure CS, CS/WQAT-4 and commercial Nafion 212 membranes in a 2 M methanol solution. As shown in Figure 7b, Nafion 212 membrane exhibited the highest current density of 535.9 mA cm^−2^ among the three membrane samples, indicating its poor methanol barrier ability due to its highly hydrophobic/hydrophilic phase separated morphology [35]. As for the pure CS membrane, it also showed a relatively high current density of 445 mA cm^−2^, while this value for the CS/WQAT-4 hybrid membrane was only 325 mA cm^−2^. This result revealed that the hybrid membrane had excellent methanol barrier ability, which is attributed to the formation of tortuous pathways against penetrating the methanol molecules in the composites.

In order to further evaluate the possibility of the practical application of our composite membranes, single DMFC was equipped and tested at 70 °C using methanol solution as the anode fuel and oxygen as the cathode gas. Figure 8a depicts the potential-current density (I-V) and the power density-current density curves of the CS/WQAT-4 composite membrane at 1 M, 2 M and 5 M methanol concentrations. From Figure 8a, the open circuit voltages (OCVs) were 0.74 V, 0.67 V and 0.64 V respectively when the methanol concentrations were 1 M, 2 M and 5 M, which showed a similar trend to that of the methanol crossover. Simultaneously, the peak power densities were 45.23, 70.26 and 59.1 mW cm^−2^ respectively at 1 M, 2 M and 3 M methanol solutions. It is obvious that the CS/WQAT-4 composite membrane exhibited a highest power density at 2 M methanol. So we further compared the single-cell performance of the CS/WQAT-4 composite membrane, pure CS and commercial Nafion 212 membranes using 2 M methanol as the anode fuel at 70 °C. Figure 8b shows that the maximum power density of Nafion 212 membrane was 79.87 mW cm^−2^, which was only slightly higher than that of the CS/WQAT-4 composite membrane (70.26 mW cm^−2^) under the same test condition. As a contrast, the DMFC equipped with the pure CS membrane output the peak power density of as low as 40.08 mW cm^−2^, which was only 57% of that of our composite membrane. The improved proton conductivity together with decreased methanol crossover may be responsible for the satisfactory fuel cell performance of our designed composite membranes.

## 4. Conclusions

In summary, super-strong proton conductor, PWA, anchored AT was prepared and directly used as a novel nanofiller employed in the chitosan matrix to fabricate composite proton exchange membranes. The mechanical strength of the composite membranes increased owing to the incorporation of uniformly-dispersed 1D WQAT, which can serve as physical cross-linking points to prohibit the rupture of polymer chains. Moreover, the ultra-strong proton conduction ability of PWA together with the interaction between positively charged CS chains and negatively charged PWA could construct effective proton transport channels with the help of 1D AT. As a result, the proton conductivity of the CS/WQAT-4% composite membrane increased by 31.8% when compared with that of the pure CS membrane at 80 °C. Besides, the methanol permeability of the composite membrane can also be remarkably decreased. The increased ionic conductivity and decreased methanol permeability lead to the increase of maximum power density for the composite electrolyte, which exhibited the value of 70.26 mW cm^−2^ at 70 °C (2 M methanol as the anode fuel), whereas the pristine membrane displayed only 40.08 mW cm^−2^.

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
