# Peer review of "Composite Proton Exchange Membranes Based on Chitosan and Phosphotungstic Acid Immobilized One-Dimensional Attapulgite for Direct Methanol Fuel Cells"

_nanomaterials, 2020, doi:10.3390/nano10091641_

Round 1

Reviewer 1 Report

The reviewer’s comments

This paper deals with the fabrication of proton-conducting composite membranes which consist of chitosan, natural attapulgite (AT), and phosphotungstic acid (PWA). The obtained membranes showed good performance on proton conductivity, and the authors conducted relevant characterizations very well. However, there are some discussions including speculations about the mechanism of proton conduction, which should be reconsidered carefully. The following comments should be addressed before publication.

Overall evaluation: major revision

Specific comments and questions

  1. line 193 on p.5: schehme.1: The amount of PWA immobilized on AT is unclear. Can the authors evaluate it?

  1. Regarding the comment 1, the particle size of WQAT is also unclear. The authors claim the dispersion of WQAT in the chitosan network, but the evidence is not provided.

  1. line 250 on p. 7: In Fig. 3, the morphologies of the membranes were provided, but the details of the structures of the composite membranes (e.g., dispersion of WQAT in the chitosan network) were not provided.

  1. line 338 on. p. 10: In Fig. 6a, the influence of the amount of WQAT in CS on proton conductivity is shown clearly. To discuss the function of PWA on AT, can the authors provide the data of proton conductivity in CS/AT? This question is related with the next comments.

  1. line 338 on. P.10: In Fig. 6b, the proton hopping path is proposed, but it looks like very speculative. Basically, the chitosan (CS) itself has good proton conductivity in this study. The extent of the contribution of PWA to proton conductivity is very unclear. Can PWA play an essential role in improving proton conductivity, according to the mechanism that the authors propose? The relevant evidence is not provided sufficiently.

  1. line 351 on p.10: Regarding the comment 5, as mentioned in the main text, the water uptake will be a governing factor for improving proton conductivity. If so, can CS/AT (not WQAT) composite satisfy the function in terms of proton conductivity? Again, the PWA function and the proposed mechanism in Fig. 6b should be discussed more carefully.

  1. line 374 on p.11: Please explain of the definition of the crossover current density.

Reviewer 2 Report

The manuscript presents an interesting investigation dealing with composite membranes for fuel cell application, in particular direct methanol fuel cell, due to the appropriate ion conductivity and proper barrier for methanol crossover.

There are just some minor questions to be considered before publication:

1. Please, include properties, methanol crossover current density and DMFC performance of a Nafion membrane for a proper comparison of the proposed composites with this benchmark membrane.

2. Explain in more detail the advantages of the proposed approach in comparison with other published strategies:

- Journal of Physical Chemistry C 118(42), 2014, 24357-24368 (DOI: 10.1021/jp5080779)

-Nanomaterials 9(9), 2019, 1292 (DOI: 10.3390/nano9091292)

- Journal of Membrane Science 599, 2020, 117858 (DOI: 10.1016/j.memsci.2020.117858)

3. Consider to include the concept of membrane selectivity to better discuss the observed enhancement of performance with the composite membrane, like in Membranes 5(4), 2015, 793-809 DOI: 10.3390/membranes5040793.
